# A cluster-randomized trial of household vs incentive-based tuberculosis contact investigation in rural South Africa: implementation reach

Eteri Machavariani[1]*, Samyra R. Cox[2], Bareng Aletta Sanny Nonyane[2], Tinatin Gvelesiani[3], Neil Martinson[4], David W. Dowdy[2], Colleen F. Hanrahan[2]

**1** School of Global Public Health, New York University, New York, New York, United States of America, **2** Department of Epidemiology, Johns Hopkins Bloomberg School of Public Health, Baltimore, Maryland, United States of America, **3** Tbilisi State Medical University, Tbilisi, Georgia, **4** Perinatal HIV Research Unit, Faculty of Health Sciences, University of the Witwatersrand Johannesburg, Johannesburg, South Africa

\* eteri.machavariani@nyu.edu

## Abstract

The World Health Organization recommends TB contact investigation in high-burden countries. We examined the implementation reach of two contact investigation strategies in South Africa**.** Kharitode TB, a cluster-randomized crossover trial, compared household- and incentive-based contact investigation in 28 clinics (July 2016-January 2020). Clinics used each strategy for 18 months (separated by a six-month "washout"). Adults recently diagnosed with TB (index participants) were enrolled. In the household-based arm, contact persons were screened and provided sputum samples at home. In the incentive-based arm, index participants distributed referral coupons to their contacts, who received a $3.50 incentive upon presenting for screening at clinics. We used mixed-effects logistic regression with random intercepts for clinics to examine factors associated with index participant enrollment and sputum collection from contact persons. In the household-based arm, 782/1,269 (61.6%) index participants consented, 1,882 contact persons were enrolled and sputum samples were collected from 988/1,882 (52.5%). In the incentive-based arm, 780/1,295 (60.2%) index participants consented, 1,940 contact persons were enrolled and sputum was collected from 1,431/1,940 (73.8%). Index participants living with HIV (adjusted odds ratio, aOR=0.56, 95% CI 0.38-0.83) or unknown HIV status (aOR=0.12, 95% CI 0.07-0.20) were less likely to participate in the study. Contact persons in the incentive-based arm were more likely to provide a sputum sample compared to those in the household-based arm (aOR=2.12, 95% CI 1.80-2.50). Regardless of the study arm, cough (aOR=2.27, 95% CI 1.87-2.77), current smoking (aOR=2.22, 95% CI 1.63-3.02), and living with HIV (aOR=1.89, 95% CI 1.36-3.62) were associated with higher likelihood of sputum collection. There were gaps in implementation reach at the stages of contacting and enrolling

**Data availability statement:** Data will be made available to researchers upon reasonable request. Requests can be directed to Johns Hopkins Bloomberg School of Public Health Epidemiology Department Chair's office at EpiChairsOffice@jh.edu.

**Funding:** The trial is funded by National Institute of Allergy and Infectious Diseases (NIAID), grant numbers: R01AI116787 to DD, R01AI147681 to DD. The funding sources had no involvement in the study design, data analysis, or manuscript preparation.

**Competing interests:** The authors have declared that no competing interests exist.

index participants, enrolling individuals with HIV, and obtaining sputum, especially among those under 18 years and household contact persons.

## Introduction

Tuberculosis (TB) is the largest global cause of morbidity and mortality caused by a single infectious agent [1]. In 2023, an estimated 10.8 million people fell ill with TB worldwide, and there were 1.25 million deaths [1]. Despite the existence of accurate and rapid diagnostic tests, only 8.2 million individuals with new TB were diagnosed and notified to healthcare authorities [1]. Addressing the gap in care for the estimated 2.6 million people with undiagnosed TB is a major focus of global TB control efforts, as delays in diagnosis and treatment continue to drive transmission. Those who live with or have close contact with someone with TB, termed contact persons, are at an increased risk of TB infection and disease [2]. Consequently, the World Health Organization (WHO) recommends TB contact investigation in high burden settings for early detection of contact persons with TB [3,4].

Contact investigation is a complex, multi-stage intervention that requires the synergy of many factors (e.g., ability to reach contact persons, availability of diagnostic tests) and stakeholders (e.g., clinicians, outreach workers, people with TB) to achieve positive outcomes in terms of TB detection and treatment initiation [5–7]. Considering this complex implementation process, effectiveness and cost-effectiveness of contact investigation have been variable within high-burden settings [2]. Measuring implementation reach – the proportion of targeted individuals effectively engaged in an intervention [8] – can provide an important understanding of barriers to effective contact investigation. However, reach as an implementation outcome is rarely investigated alongside effectiveness. The aim of this analysis is to examine the implementation reach, defined as the proportion of individuals completing each step in the contact investigation cascade, in two contact investigation strategies: a traditional household-based investigation and a novel incentive-based approach within the context of a cluster-randomized trial in South Africa [9–11].

## Methods

### Study design

This is a secondary analysis of a cluster-randomized trial (Kharitode TB) of TB case finding strategies in Limpopo Province, South Africa from July 18, 2016 to January 17, 2020. The trial aimed to compare the effectiveness of facility-based screening to household and incentive-based strategies of contact investigation, focusing on the number of incident TB diagnoses and treatment initiation as the primary outcome. A more detailed description of the trial design is published elsewhere [9,10]. Briefly, a total of 56 public-sector primary care clinics were randomized 1:1 to either facility-based screening or contact investigation. Clinics implementing a contact investigation strategy were further divided into those using a household-based strategy versus those using a novel incentive-based approach. This analysis focuses only on the contact investigation arm.

## Local context

South Africa is a high TB burden country, with an estimated 513 incident cases per 100,000 in 2021 [12]. The population is divided relatively evenly between urban and rural areas, with prevalent seasonal work-related migration, which makes contact investigation challenging and contributes to TB transmission patterns [13]. The Kharitode TB trial was carried out in two rural districts of Limpopo Province – Vhembe and Waterberg. Although the population density in these rural districts is low (<50/km$^2$), TB prevalence and incidence remain high, reflecting, in part, socioeconomic conditions, malnutrition, and difficulties in healthcare access due to long distances and transportation barriers [14,15].

## Contact investigation strategies and eligibility criteria

Contact investigation was carried out in 28 of the 56 study clinics. Among these, 14 clinics initially implemented the household-based strategy and 14 implemented the incentive-based strategy. Clinics were randomized to one strategy for 18 months and switched to the opposite strategy for 18 months after a 6-month "washout" period during which neither strategy was employed. The present analysis includes data from both 18-month intervention periods. All patients diagnosed with TB and initiated on TB treatment at participating clinics within two months of study launch (potential index participants), as well as their respective contact persons, were eligible for enrollment, irrespective of age and mode of diagnosis. The study team identified index participants for recruitment using clinic TB treatment registers (both electronic and paper).

In the household-based contact investigation arm, the study team made three phone call attempts to reach each potential index participant on varying days and times to obtain verbal consent. If the person was unreachable by phone, the study team conducted a household visit. Following index participant enrollment, a date and time were set for a household visit to screen contact persons for TB. At the household, each consenting contact person was screened using the WHO four-symptom screen (presence of any of the following: current cough, fever, weight loss, or night sweats) [16], and an attempt was made to obtain a sputum sample regardless of symptoms. The study team made up to three visits to enroll and screen all household contact persons. Participants were compensated for time spent for research purposes.

In the incentive-based contact investigation arm, procedures for recruiting and enrolling index participants were similar to those in the household-based arm. Each index participant received a set of 10 coupons to be distributed among their household members, friends, or coworkers. The coupon contained the working hours and the address of the clinic along with the unique identification code of the index participant. Each contact person presenting to the clinic for TB screening within two months of index participant enrollment was given 50 South African Rand (ZAR), approximately equal to 3.50 USD. Index participants were reimbursed 20 ZAR (1.50 USD) for each contact person presenting for screening, and 100 ZAR (7.00 USD) for each contact person diagnosed with TB. Upon presenting to clinics, contact persons were screened using the WHO four-symptom screen [16] and sputum samples were obtained.

In both contact investigation arms, sputum samples were transported by study staff to study clinics, and afterwards to the National Health Laboratory Services via routine daily standard of care transport routes for Xpert MTB/RIF testing [17]. Results were accessed via the electronic laboratory information system, and study teams then informed participants of their Xpert MTB/RIF results during a household visit or telephonically. Contact investigation process for both arms is presented in Fig 1.

Data sources included a combination of paper and electronic logs to document completion of each step in the implementation cascade for each contact investigation strategy. Potential index participant characteristics, including age, sex, and HIV status, were obtained from clinic datasets, whereas contact person characteristics were self-reported through surveys.

## Study aims

The aim of this analysis was to evaluate the implementation reach of contact investigation strategies by characterizing the proportion of index participants enrolled and assessing differences in contact person enrollment and sputum sample

PLOS Global Public Health

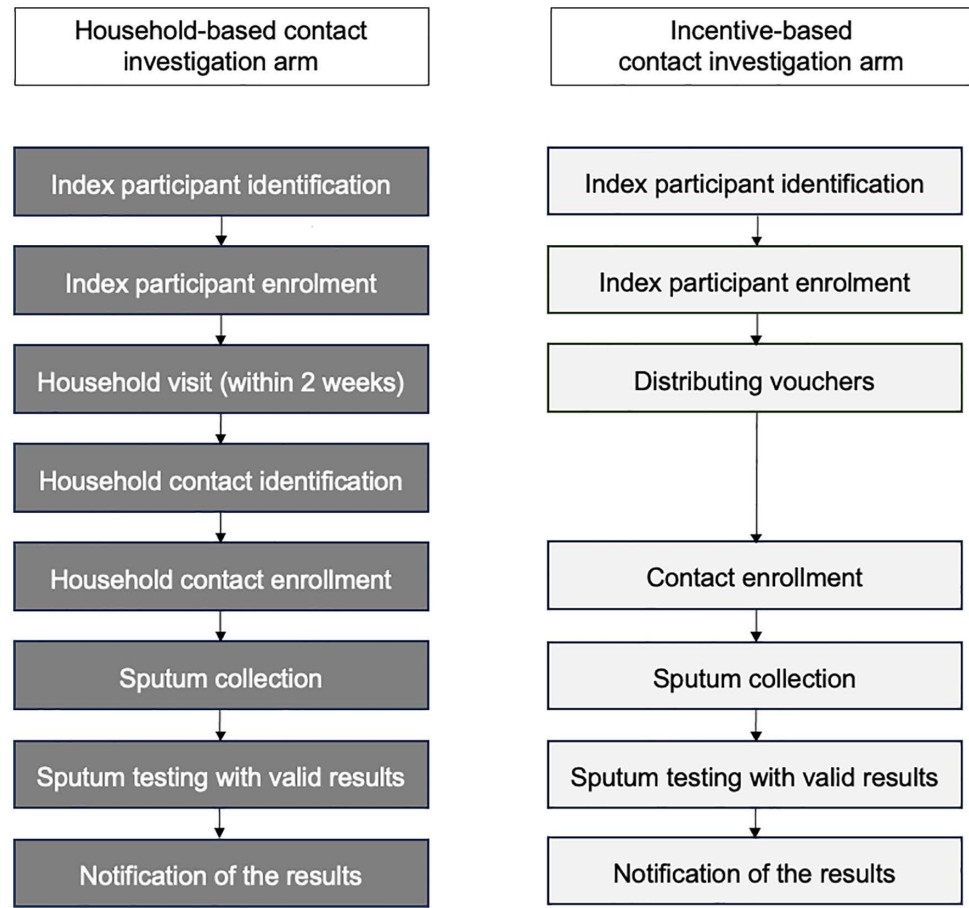

**Fig 1. Contact investigation process flow-chart for household- and incentive-based strategies.**

collection between the household-based and incentive-based contact investigation strategies. Additionally, we aimed to explore the sociodemographic and clinical characteristics associated with successful participation among index participants and contact persons.

## Implementation reach outcomes

In this analysis, we operationalized implementation reach by measuring the proportion of individuals completing each step in the contact investigation cascade. Specifically, we assessed the proportion of eligible individuals diagnosed with TB who were enrolled in the contact investigation program within two months of diagnosis (index participant enrollment), the proportion of enrolled index participants for whom at least one eligible contact person was screened for TB (contact screening), the proportion of contact persons who provided a sputum sample for testing (sputum sample collection), the proportion of collected samples that were tested with a valid result (sample testing), and the proportion of contact persons who were successfully notified of their Xpert MTB/RIF results (result notification).

## Data analysis

To characterize implementation reach components, we first calculated the proportion of index participants enrolled within two months of TB diagnosis, taking as the denominator the total number of potential index participants identified, pooled

**Global Public Health**

between the two arms. Data from both 18-month intervention periods were combined for analysis, as this secondary analysis was descriptive and aimed to evaluate overall implementation reach rather than intervention effects.

Next, we calculated the proportion of enrolled index participants for whom at least one contact person was screened. For these enrolled contact persons, we calculated the percentages: (1) whose sputum samples were collected, (2) whose samples were tested, (3) whose samples were tested with valid test results, and (4) who were notified of their Xpert results.

We explored the characteristics of populations reached through the intervention using adjusted mixed-effects logistic regression models with a random intercept for clinic to account for clustering at the clinic level. The primary outcome was consent to participate, and independent variables included sex, age, study district, and HIV status. We also modeled, among consenting index participants, the likelihood that at least one of their contacts enrolled in the study, separately for each arm. For contact persons, the outcome was provision of a sputum sample, and covariates included sex, age, study district, self-reported symptoms, smoking status, and HIV status. We included study arm as a fixed effect in the mixed-effects logistic regression models to directly assess differences between arms while accounting for clustering at the clinic level. R statistical software version 4.4.1 was used for the analysis [18].

The de-identified data were accessed for research purposes between September 12, 2023 and May 3, 2025.

## Ethical considerations

Independent ethics review was conducted by the Human Research Ethics Committee at the University of the Witwatersrand in South Africa (#00001223). The Institutional Review Board at the Johns Hopkins Bloomberg School of Public Health provided authorization to rely on the Human Research Ethics Committee at the University of the Witwatersrand for review and continuous oversight of this trial. Written informed consent was obtained from all participants and the consent procedures were approved by the University of the Witwatersrand Human Research Ethics Committee. The trial is registered with ClinicalTrials.gov (NCT02808507).

## Results

Overall, the study team identified 2,563 people diagnosed with TB as potential index participants in the contact investigation arm, of whom 1,562 (60.9%) were enrolled in either study arm during the study period. Enrollment of potential index participants was similar across arms (61.6% and 60.3% in the household-based and incentive-based arms, respectively), though formal comparisons accounting for clustering are presented in the mixed-effects logistic regression models below. Exhaustion of contact attempts, inability to obtain informed consent (due to death, critical illness, unavailability, or relocation), and refusal to participate were the leading reasons for failure to enroll index participants. Exhaustion of contact attempts was the most common cause of enrollment failure for 439 (17.1%) potential index participants, including 189 (14.9%) in the household-based and 250 (19.3%) in the incentive-based arm. A total of 190 (7.4%) of potential index participants declined to participate, with 116 (9.1%) and 74 (5.7%) in the household-based and incentive-based arms, respectively. Enrollment details are provided in S1 Table in S1 File.

Table 1 presents the characteristics of potential index participants who were successfully reached by the study team and either consented to or declined participation. The sample was similar across intervention arms, with the exception of a slightly larger proportion enrolling in the incentive-based arm (91.3% versus 87.1%). In crude models, participants in the Waterberg District were less likely than those living in the Vhembe District to enroll in the study, though this association did not persist following the adjustment for other variables. In the adjusted model, index participants were more likely to enroll in the incentive-based arm (aOR=1.58, 95% CI 1.14-2.17). Those individuals living with HIV diagnosis (aOR=0.56, 95% CI 0.38-0.83) and unknown HIV status (aOR=0.12, 95% CI 0.07-0.20) were less likely to enroll as opposed to individuals not living with HIV (Table 2). There was no significant interaction by HIV status and study arm.

**Table 1. Characteristics of potential index participants reached by the study team who consented to or declined participation, by study arm.**

| | Household-based arm (N = 898) | Incentive-based arm (N = 854) | Total (N = 1752) |
|---|---|---|---|
| Female | 371 (41.3) | 361 (42.3) | 732 (41.8) |
| Age, years [Mean (SD)] | 39.51 (14.92) | 38.95 (15.91) | 39.24 (15.41) |
| Age ≥ 18 years | 850 (95.0) | 786 (92.6) | 1636 (93.8) |
| Vhembe district | 390 (43.4) | 365 (42.7) | 755 (43.1) |
| Waterberg district | 508 (56.6) | 489 (57.3) | 997 (56.9) |
| HIV negative | 323 (36.0) | 314 (36.8) | 637 (36.4) |
| HIV positive | 513 (57.1) | 493 (57.7) | 1006 (57.4) |
| HIV status unknown | 62 (6.9) | 47 (5.5) | 109 (6.2) |
| Enrolled in the study | 782 (87.1) | 780 (91.3) | 1562 (89.2) |

**Table 2. Characteristics associated with enrolling in the study among potential index participants (N = 1744).**

| | Crude Logistic Regression | | Adjusted Logistic Regression | |
|---|---|---|---|---|
| | OR | 95% CI | aOR | 95% CI |
| Female | Ref | | Ref | |
| Male | 0.90 | 0.66 – 1.23 | 0.89 | 0.64 – 1.23 |
| Age < 18 years | Ref | | | |
| Age ≥ 18 years | 0.65 | 0.31 – 1.36 | 0.56 | 0.26 – 1.26 |
| Household-based arm | Ref | | Ref | |
| Incentive-based arm | 1.65 | 1.21 – 2.26 | 1.58 | 1.14 – 2.17 |
| Vhembe | Ref | | Ref | |
| Waterberg | 0.77 | 0.51 – 1.16 | 0.82 | 0.53 – 1.27 |
| HIV negative | Ref | | Ref | |
| HIV positive | 0.55 | 0.37 – 0.80 | 0.56 | 0.38 – 0.83 |
| HIV status unknown | 0.12 | 0.07 – 0.20 | 0.12 | 0.07 – 0.20 |

Mixed-effects logistic regression with a random intercept for clinic was adjusted for sex, age, arm, district, and HIV status.

OR=odds ratio; aOR=adjusted odds ratio; CI=confidence interval.

Fig 2 represents the implementation reach cascade for both household and incentive-based contact investigation strategies. In the household-based contact investigation arm, 782 (61.6%) of the 1,269 identified people with TB consented to participate. Households were visited for 766 (60.4%) enrolled index participants, and at least one contact person was screened in 519 (40.9%) households with an identified index participant, i.e., per index participant. Among contacts, 1,882 individuals were enrolled from these 519 households. Sputum specimens were obtained from 988 (52.5%) of the enrolled contact persons, 978 specimens (52.0%) were tested, and 898 (47.7%) had valid results. Finally, 953 (50.6%) contact persons were notified of their Xpert MTB/RIF results. Restricting to only those aged 18 or older, sputum was collected from 615 (62.8%) of the 980 enrolled adult contact persons.

In the incentive-based contact investigation arm, a total of 780 (60.2%) of 1,295 identified index participants were enrolled; all enrolled participants received coupons. In this arm, at least one contact person presented to a clinic for screening for 349 (26.9%) index participants. Sputum specimens were collected from 1,431 (73.8%) contact persons, 1,405 specimens (72.4%) were tested, and 1,342 (69.2%) had valid results. The study team successfully notified 1,391 (71.0%) participants of their Xpert MTB/RIF test results. Overall, the screening, testing, and notification of the results were completed for 953/1,882 (50.6%) contact persons in the household-based arm and 1,391/1,940 (71.0%) in the incentive-based arm.

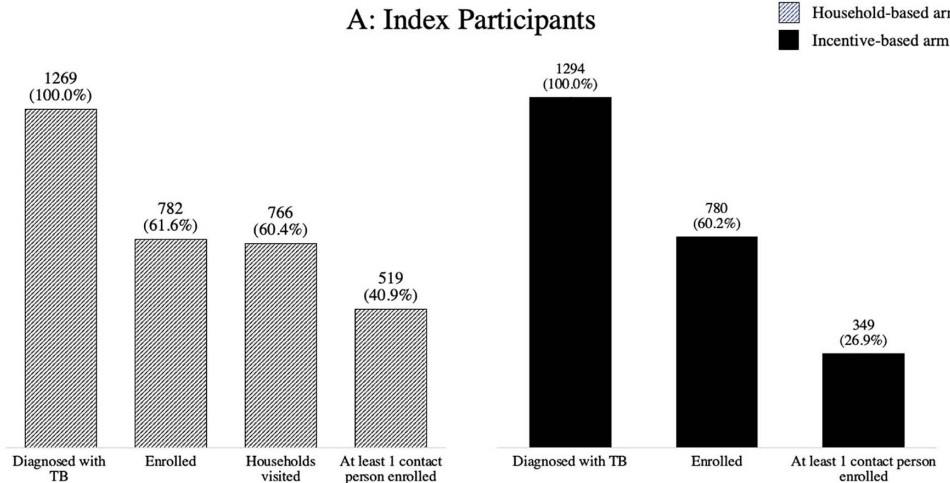

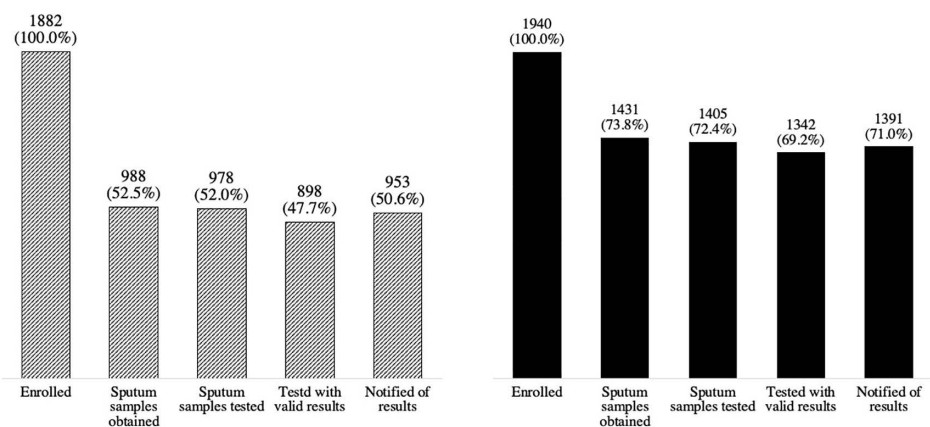

**Fig 2. Reach cascade of the implementation of household- and incentive-based contact investigation strategies.**

Male sex was negatively associated with contact enrollment among index participants in both household- (aOR 0.60, 95% CI 0.43-0.83) and incentive-based arms (aOR 0.65, 95% CI 0.48-0.89). Residence in the Waterberg study district (aOR 0.34, 95% CI 0.23-0.48), in the household-based arm, and being younger than 18 in the incentive-based arm (aOR 0.34, 95% CI 0.18-0.62) were also associated with lower odds of enrolling at least one contact person (S2 Table in S1 File).

Characteristics of contact persons stratified by sputum collection status in each arm are presented in Table 3. In the adjusted model, contact persons in the incentive-based arm were more likely to provide a sputum sample compared to those in the household-based arm (aOR=2.12, 95% CI 1.80-2.50). Across both arms, individuals aged 18 years or older (aOR=2.93, 95% CI 2.33-3.68), those presenting with cough (aOR=2.27, 95% CI 1.87-2.77), current smokers (aOR=2.22, 95% CI 1.63-3.02) and those living with HIV (aOR=1.89, 95% CI 1.36-2.62) were more likely to provide a sputum sample. Conversely, residing in Waterberg District was associated with lower odds of sputum provision (aOR=0.28, 95% CI 0.22-0.35) (Table 4). We assessed potential effect modification by study arm for selected covariates (sex, HIV status, and study district). No statistically significant interactions were observed.

**Table 3. Characteristics of contact persons.**

| | Household-based Arm | | | Incentive-based Arm | | |
|---|---|---|---|---|---|---|
| | Sputum not collected | Sputum collected | Total | Sputum not collected | Sputum collected | Total |
| | (N = 902) | (N = 988) | (N = 1890) | (N = 511) | (N = 1431) | (N = 1942) |
| Female | 553 (61.3) | 607 (61.7) | 1160 (61.5) | 306 (59.9) | 814 (58.1) | 1120 (58.6) |
| Age ≥ 18 years | 365 (41.9) | 615 (62.5) | 980 (52.8) | 204 (41.4) | 1088 (77.7) | 1292 (68.3) |
| Vhembe | 369 (40.9) | 612 (62.2) | 981 (52.0) | 162 (31.7) | 728 (52.0) | 890 (46.6) |
| Waterberg | 533 (59.1) | 372 (37.8) | 905 (48.0) | 349 (68.3) | 672 (48.0) | 1021 (53.4) |
| Tuberculosis Symptoms: | | | | | | |
| Cough | 138 (15.3) | 278 (28.3) | 416 (22.1) | 108 (21.1) | 472 (33.7) | 580 (30.0) |
| Fever | 57 (6.5) | 66 (6.7) | 123 (6.5) | 32 (6.3) | 137 (9.8) | 169 (8.8) |
| Weight loss | 46 (5.1) | 59 (6.0) | 105 (5.6) | 32 (6.3) | 168 (12.0) | 200 (10.5) |
| Night sweats | 70 (7.8) | 89 (9.0) | 159 (8.4) | 47 (9.2) | 249 (17.8) | 296 (15.5) |
| Other * | 120 (13.4) | 184 (18.7) | 304 (16.2) | 91 (17.2) | 353 (25.3) | 444 (23.3) |
| Smoking Status | | | | | | |
| Never smoker | 825 (92.4) | 843 (85.7) | 1668 (88.9) | 467 (91.7) | 989 (70.7) | 1456 (76.3) |
| Yes | 45 (5.0) | 104 (10.6) | 149 (7.9) | 30 (5.9) | 321 (22.9) | 351 (18.4) |
| Not currently, but formerly | 22 (2.5) | 34 (3.5) | 56 (3.0) | 10 (2.0) | 84 (6.0) | 94 (4.9) |
| Declined to disclose | 1 (0.1) | 3 (0.3) | 4 (0.2) | 2 (0.4) | 5 (0.4) | 7 (0.4) |
| HIV negative | 702 (77.8) | 778 (79.1) | 1480 (78.5) | 273 (53.6) | 779 (55.7) | 1052 (55.1) |
| HIV positive | 42 (4.7) | 53 (5.4) | 95 (5.0) | 21 (4.1) | 185 (13.2) | 20 (10.8) |
| Declined to disclose/unknown | 158 (17.5) | 153 (15.5) | 311 (16.5) | 217 (42.5) | 436 (31.1) | 653 (34.2) |

* Other symptoms included: chest pain, pain in another part of the body, skin problems, stomach or intestinal problems, genital or urinary problems.

## Discussion

Improving implementation of TB contact investigation is a key priority in increasing the effectiveness of this intervention; however, only a handful of studies have reported implementation outcomes in high-burden settings [19–23]. This analysis of two contact investigation strategies – household-based and incentive-based – found a significant difference in the proportion of index participants whose contacts were screened for TB. Fewer than half of *potential* index participants in either arm had at least one contact screened for TB, with lower screening observed in the incentive-based arm (27%) compared to the household-based arm (41%). Among enrolled contact persons, implementation reach was higher in the incentive-based arm, with over two-thirds of all contact persons enrolled notified of their Xpert MTB/RIF results as opposed to only half in the household-based arm.

We found that fewer than two thirds of potential index participants were enrolled in the study. The main reason for the failure to enroll index participant was exhaustion of attempts to contact them – the study team was unable to reach these participants, either by telephone or household visits. The lack of mobile phones, people changing their mobile phone numbers, short-term work mobility, and difficulties in finding houses without addresses may be among contributing factors to enrolment failure [10,24]. Lack of mobile phones has been reported as a barrier to TB service implementation in India [25], while malfunctioning of mobile devices and phone-sharing practices were documented as barriers in Uganda [20].

We found that those living with HIV or those whose HIV status was unknown were less likely to consent to study participation in either arm, compared to individuals without HIV. One potential explanation behind this is the well-documented stigma associated with HIV and TB in South Africa, where being diagnosed with HIV and TB has severe social consequences [26]. HIV and TB stigma commonly intersect in these settings: it has been shown that those diagnosed with HIV

**Table 4. Characteristics associated with sputum collection status from contact persons, stratified by arm.**

| | Crude Logistic Regression | | Adjusted Logistic Regression | |
|---|---|---|---|---|
| | OR | CI | aOR | CI |
| Household-based arm | Ref | | Ref | |
| Incentive-based arm | 2.67 | 2.31 – 3.08 | 2.12 | 1.80 – 2.50 |
| Female | Ref | | Ref | |
| Male | 1.05 | 0.91 – 1.20 | 1.05 | 0.88 – 1.25 |
| Age < 18 years | Ref | | Ref | |
| Age ≥ 18 years | 3.99 | 3.44 – 4.64 | 2.93 | 2.33 – 3.68 |
| Vhembe district | Ref | | Ref | |
| Waterberg district | 0.42 | 0.33 – 0.53 | 0.28 | 0.22 – 0.35 |
| No reported cough | Ref | | Ref | |
| Cough | 2.36 | 1.99 – 2.79 | 2.27 | 1.87 – 2.77 |
| No reported Fever | Ref | | Ref | |
| Fever | 1.69 | 1.29 – 2.21 | 0.96 | 0.69 – 1.33 |
| No reported weight loss | Ref | | Ref | |
| Weight loss | 2.28 | 1.73 – 3.00 | 1.11 | 0.80 – 1.54 |
| No reporter night sweats | Ref | | Ref | |
| Night sweats | 2.00 | 1.59 – 2.52 | 1.00 | 0.76 – 1.32 |
| Never smoker † | Ref | | Ref | |
| Current smoker | 5.02 | 3.84 – 6.57 | 2.22 | 1.63 – 3.02 |
| Not currently, but formerly | 3.05 | 2.02 – 4.59 | 1.37 | 0.88 – 2.13 |
| HIV negative | Ref | | Ref | |
| HIV positive | 2.45 | 1.82 – 3.30 | 1.89 | 1.36 – 2.62 |
| Declined to disclose/unknown | 1.02 | 0.87 – 1.20 | 1.12 | 0.93 – 1.35 |

Mixed-effects logistic regression with a random intercept for clinic was adjusted for arm, sex, age, district, tuberculosis symptoms, smoking and HIV status.

† The category encompasses never smokers and those who declined to disclose their smoking status.

OR = odds ratio; aOR = adjusted odds ratio; CI = confidence interval.

experience higher levels of TB stigma and that the two are linked to poor care-seeking behavior and sub-optimal TB treatment outcomes [27–31]. We also observed that contact persons with HIV in the incentive-based arm were more likely to provide sputum samples, as opposed to those in the household arm. Potential explanations include that contact persons with HIV may have had more advanced illness or symptoms, increasing their likelihood of providing a sample, and that the incentive-based strategy, conducted at the clinic, may have offered greater privacy than household visits, reducing fears of HIV- and TB-related stigma and encouraging participation.

The analysis also highlights challenges in sputum sample collection during TB contact investigation, especially in the household-based arm. Four-fifths of the contact persons in the incentive-based arm were able to produce sputum, similar to an active case finding strategy in Peru, where sputum collection fidelity was over 85% in a clinical setting, but comparatively lower than a study carried out in Capricorn, where sputum samples were successfully obtained from 92% of the adult population [21,32]. In the household-based arm, however, just over half of the participants were able to provide sputum samples. One reason behind this discrepancy between arms may be the cadre of the worker supervising sputum collection and/or the location – in the household arm, sputum was collected at the household by a lay healthcare worker, whereas in the incentive-based arm, it was collected by an experienced TB

healthcare worker at the clinic. Finding safe spaces to collect sputum samples in households may have proven to be challenging. In a mixed-methods study in Uganda, lay health workers were able to obtain sputum samples from only 39% of household contact persons, and the inability to find a secluded space outdoors for sputum collection was cited as one of the impeding factors [33].

Another reason explaining lower rates of sputum collection in households may be that there was a larger proportion of children among contact persons in this arm compared to the incentive-based arm. We observed that those aged 18 years or older were almost three times more likely to produce sputum samples compared to those under 18. Children have difficulties in producing a sufficient amount of sputum compared to adults which makes sputum collection challenging [34], while they are at a heightened risk of developing TB disease as contact persons. Alternatives to sputum samples for TB diagnosis, such as oral swabs [35] could be of utility in TB contact investigation for children and adults who cannot produce sputum, though current diagnostic accuracy of these strategies is sub-optimal. However, this does not fully account for the lower sputum collection rates in the household arm, as even after adjusting the analysis for age and other variables, those in the incentive-based arm were twice as likely to produce samples for sputum collection as their peers in the household-based arm.

We did not measure the implementation reach of coupon distribution by participants. However, it is important to note that there was a potential loss of participation at this stage. In the household-based arm, the majority of *enrolled* index participants were linked to at least one contact person, whereas in the incentive-based arm, less than half of enrolled index participants had a linked contact person. Multiple factors may explain this discrepancy. First, some index participants may not have fully understood the coupon system. Second, index participants may have failed to distribute coupons among their contact persons because of the stigma of revealing their TB status [36,37]. Finally, contact persons may have faced difficulty in presenting to clinics due to transportation problems, distance, or insufficient cash incentives. Even though cash incentives have been shown to be effective in enrollment, engagement, and retention in various health programs [38], and in TB related programs in particular [39,40], the amount of incentives plays a crucial role and might influence the uptake of a healthcare intervention in a given context. Studies in Uganda showed, however, that the amount close to the cash incentive provided in this trial (1.00-3.50 USD) was sufficient to motivate patients to participate in TB programs [41,42]. However, the effectiveness of such incentives may vary depending on the relative household income, and economic context in each country.

Several limitations of this study should be noted. First, contact investigations were carried out by dedicated study staff, thus the measures of implementation reach reported here may not be generalizable to more routine implementation conditions, where healthcare workers may have different qualifications or experience. Second, we were not able to measure the reach of coupon distribution (e.g., whether index participants indeed distributed all 10 coupons to their close contacts as instructed); however, our analysis of the proportion of coupons returned to the clinic by contact person lends some insight into this step of incentive-based contact investigation. Third, antiretroviral therapy (ART) status of index participants and contact persons with HIV was not known and it was not possible to understand if ART engagement affected study participation differentially. Finally, the addition of qualitative interviews after contact investigation with index participants, contacts could shed more light on specific provider-, participant-, and contextual barriers.

## Conclusions

This study provides valuable insights into the implementation reach of TB contact investigation strategies. Difficulties in contacting and enrolling index participants, lower participation among individuals with HIV, and challenges in contact screening and sputum collection – particularly among children – limited the implementation reach of both household- and incentive-based TB contact investigation strategies. These findings characterize operational gaps in the contact investigation cascade that need to be addressed to achieve the maximum impact of the intervention.

## Supporting information

**S1 File.** **S1 Table**. Enrollment of index participants (people diagnosed with TB) in each arm. **S2 Table**. Index participant characteristics associated with enrolling at least one contact person.
(DOCX)

## Author contributions

**Conceptualization:** Eteri Machavariani, Neil Martinson, David W. Dowdy, Colleen R. Hanrahan.

**Data curation:** David W. Dowdy, Colleen R. Hanrahan.

**Formal analysis:** Eteri Machavariani, Colleen R. Hanrahan.

**Funding acquisition:** David W. Dowdy.

**Investigation:** Eteri Machavariani, Neil Martinson, David W. Dowdy, Colleen R. Hanrahan.

**Methodology:** Samyra R. Cox, Bareng Aletta Sanny Nonyane, Neil Martinson, David W. Dowdy.

**Project administration:** Colleen R. Hanrahan.

**Resources:** Neil Martinson, Colleen R. Hanrahan.

**Supervision:** Neil Martinson, David W. Dowdy, Colleen R. Hanrahan.

**Validation:** Bareng Aletta Sanny Nonyane, Tinatin Gvelesiani.

**Visualization:** Eteri Machavariani.

**Writing – original draft:** Eteri Machavariani, Samyra R. Cox.

**Writing – review & editing:** Eteri Machavariani, Samyra R. Cox, Bareng Aletta Sanny Nonyane, Tinatin Gvelesiani, David W. Dowdy, Colleen R. Hanrahan.

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
