## [Decision Letter · Decision Letter 0]

2 Sep 2025

PGPH-D-25-01439

A cluster-randomized trial of household vs incentive-based tuberculosis contact investigation in rural South Africa: implementation reach outcomes

Dear Dr. Machavariani,

Thank you for submitting your manuscript to PLOS Global Public Health. After careful consideration, we feel that it has merit but does not fully meet PLOS Global Public Health’s publication criteria as it currently stands. Therefore, we invite you to submit a revised version of the manuscript that addresses the points raised during the review process.

We look forward to receiving your revised manuscript.

Kind regards,

Joseph Baruch Baluku, MMed

Academic Editor

Journal Requirements:

1. In the ethics statement in the Methods, you have specified that verbal consent was obtained. Please provide additional details regarding how this consent was documented and witnessed, and state whether this was approved by the IRB

2. We have amended your Competing Interest statement to comply with journal style. We kindly ask that you double check the statement and let us know if anything is incorrect.

3. In the online submission form, you indicated that “Data will be made available upon reasonable request.”

a. In a public repository,

b. Within the manuscript itself, or

c. Uploaded as supplementary information.

4. Your current Financial Disclosure states, “No financial interests to disclose.”. However, your funding information on the submission form indicates that you received funding from “National Institute of Allergy and Infectious Diseases with grant number R01AI116787 and R01AI147681”. Please indicate by return email the full and correct funding information for your study and confirm the order in which funding contributions should appear. Please be sure to indicate whether the funders played any role in the study design, data collection and analysis, decision to publish, or preparation of the manuscript.

Additional Editor Comments (if provided):

Reviewer #1:

Reviewer #2:

Reviewer #3:

Reviewers' comments:

Reviewer's Responses to Questions

**Comments to the Author**

1. Does this manuscript meet PLOS Global Public Health’s publication criteria?

Reviewer #1: Yes

Reviewer #2: No

Reviewer #3: Yes

2. Has the statistical analysis been performed appropriately and rigorously?

Reviewer #1: Yes

Reviewer #2: No

Reviewer #3: Yes

3. Have the authors made all data underlying the findings in their manuscript fully available (please refer to the Data Availability Statement at the start of the manuscript PDF file)?

Reviewer #1: No

Reviewer #2: No

Reviewer #3: Yes

4. Is the manuscript presented in an intelligible fashion and written in standard English?

Reviewer #1: Yes

Reviewer #2: Yes

Reviewer #3: Yes

Reviewer #1: Thank you for this interesting and relevant study, as contact tracing of people with TB represents a large barrier to achieving testing and treatment goals, and the use of incentives in TB services is not well understood.

While the manuscript is well-written and the analysis appears to be sound, my major issue is to do with incomplete sentences, specifically Methods, line 94 and Conclusion: page 22, line 385 ends with "These findings". As a result of not knowing what the missing information entails, I would request the authors resubmit and consider proof-reading before doing so.

Minor comments:

The secondary analysis is restricted to data between July 2016 to January 2018, representing 18 months, but the intervention describes 18 months per intervention in all clinics including a 6 month period of no intervention. This would imply 42 months ini total. Did the analysis use the first or second 18 month intervention? Please clarify which period is being analysed or if this has been misunderstood.

The authors describe making home visits when unable to reach participants by phone as a method of overcoming this barrier, however limitations related to relying on contact by mobile phones and possible reasons for this are discussed (line 305-307). Were these reasons documented when study staff were unable to reach participants by phone? Were any other methods for reaching participants considered such as making use of TB tracer teams or community health workers? As these are important aspects of retention in care.

Concerning lower participation among people living with HIV, is it known whether or not these people were already engaged in chronic ART care and may have already had contacts screened for TB?

Reviewer #2: This study is a an embedded study from a cluster randomized trial. The cluster randomized trial randomized 56 study clinics to facility based screening or contact investigation in South Africa. Here, they consider the 28 clinics randomized to contact investigation. They compare 14 with traditional contact investigation to 14 with incentive based contact investigation on the TB cascade of diagnosis. The comparison lasted for 18 months, then after a 6 month washout period, facilities switched interventions. They compared the proportion of individuals completing each step of the contact investigation cascade.

All statistical tests should take into account the clustering by facility, and possibly clustering by index patient. Observations are not independent.

It is not clear how the crossover nature of the data was dealt with. Please clarify.

1. Please justify using the pooled number of individuals for the denominator of the proportion of index participants enrolled within 2 months of TB dx.

2. The proportion of index participants for whom at least one contact person was screened was compared between arms via chi-squared test - this is not appropriate. Clustering must be accounted for. Similarly, for all the other proportions.

3. There may also be clustering by index patient for the other proportions based on contacts. This clustering should be accounted for.

4. For the multivariable models used to explore index participant characteristics -- mixed logistic models should be used to account for clustering by clinic.

5. For the multivariable models used to explore contact participant characteristics -- models that allow for clustering by clinic and by index patient should be used.

6. In all tables (including supplemental ones), just present a CI. No need to also present a p value.

7. In the discussion, it is emphasized that fewer than half of potential index patients had at least one contact screened and that this was lower in the incentive arm than the household based arm. However, this seems to ignore the fact that the incentive based arm did a lot better at continuing to the end - overall 0.75=953/1269 contacts notified of results per potential index cases in the household arm vs. 1.07=1391/1294 in the incentive arm. This seems pretty important yet is not discussed - why?

8. The Conclusions section is cutoff at the end.

9. Please specify which variables were included in the models in a footnote in Suppl Table 2.

10. Table 3 seems a bit repetitive with Table 4. Remove the p-values from Table 3 - these comparisons are presented in Table 4.

11. For Table 4 - is it interesting to compare the characteristics by arm?

12. Everywhere - instead of univariate and multivariable use "Crude" and "Adjusted".

Reviewer #3: This well written manuscript contains important information that could guide the Household Contact TB programs in SA. Here are minor comments for consideration.

1. Were all participants reimbursed? Was the reimbursement separate from the incentive?

2. What was the incentive? This is not mentioned in the manuscript. It may have been described elsewhere but it is also worthwhile to briefly describe it in this manuscript and then provide a citation.

3.Line 152 indicates that some diagnostic outcomes were assessed. However, i have not seen any diagnostic outcomes reported in the results.

4. Were the people living with HIV on ART? Sometimes, PLHW on ART are unable to produce sputum due to their generally improved health. Table 4 indicates a strong association between being HIV positive and sputum collection/production and i was just wondering how this association would have looked like with PLWH on established ART vs newly on ART.

5. Table 4 also shows a strong association with cough and sputum collection and no association with other TB symptoms. What would be the implications of subclinical TB on household contact programs if there were no TB symptoms?

6. Table 1: What are the "other" TB symptoms? elaborate in a footnote

7. Were the HIV status and smoking status self-reported? if yes, what are the limitations for these?

8. Include a citation for R software

9. Include a reference number for Wits HREC ethics approval.

**Do you want your identity to be public for this peer review?** For information about this choice, including consent withdrawal, please see our Privacy Policy

Reviewer #1: No

Reviewer #2: No

Reviewer #3: No

---

## [Editor Report · Decision Letter 1]

18 Nov 2025

A cluster-randomized trial of household vs incentive-based tuberculosis contact investigation in rural South Africa: implementation reach

PGPH-D-25-01439R1

Dear Dr Machavariani,

We are pleased to inform you that your manuscript 'A cluster-randomized trial of household vs incentive-based tuberculosis contact investigation in rural South Africa: implementation reach' has been provisionally accepted for publication in PLOS Global Public Health.

Best regards,

Joseph Baruch Baluku, MMed

Academic Editor